# The injectable contraceptives depot medroxyprogesterone acetate and norethisterone enanthate substantially and differentially decrease testosterone and sex hormone binding globulin levels: A secondary study from the WHICH randomized clinical trial

Chanel Avenant[1]◉, Mandisa Singata-Madliki[2]◉, Alexis J. Bick[1], Donita Africander[3], Yusentha Balakrishna[4], Karl-Heinz Storbeck◉[3], Johnson M. Moliki[1], Sigcinile Dlamini[1], Salndave Skosana[1†], Jenni Smit[5], Mags Beksinska[5], Ivana Beesham[5], Ishen Seocharan[4], Joanne Batting[2], George J. Hofmeyr[2,6,7‡], Janet P. Hapgood◉[1,8‡]*

1 Department of Molecular and Cell Biology, University of Cape Town, Cape Town, South Africa, 2 Effective Care Research Unit, Eastern Cape Department of Health/Universities of the Witwatersrand and Fort Hare, East London, South Africa, 3 Department of Biochemistry, Stellenbosch University, Stellenbosch, South Africa, 4 Biostatistics Research Unit, South African Medical Research Council, Durban, South Africa, 5 Wits MRU (MatCH Research Unit), Department of Obstetrics and Gynecology, Faculty of Health Sciences, University of the Witwatersrand, Durban, South Africa, 6 Walter Sisulu University, East London, South Africa, 7 Department of Obstetrics and Gynecology, University of Botswana, Gaborone, Botswana, 8 Institute of Infectious Disease and Molecular Medicine, University of Cape Town, Cape Town, South Africa

◉ These authors contributed equally to this work.
† Deceased.
‡ GJH and JPH also contributed equally to this work.
* Janet.Hapgood@uct.ac.za

## Abstract

HIV acquisition risk with norethisterone (NET) enanthate (NET-EN) is reportedly less than for depo-medroxyprogesterone acetate intramuscular (DMPA-IM). We investigated the effects of these progestin-only injectable contraceptives on serum testosterone and sex hormone binding globulin (SHBG) levels, since these may play a role in sexual behavior and HIV acquisition. The open-label WHICH clinical trial, conducted at two sites in South Africa from 2018–2019, randomized HIV-negative women aged 18–40 years to 150 mg DMPA-IM 12-weekly (n = 262) or 200 mg NET-EN 8-weekly (n = 259). We measured testosterone by UHPLC-MS/MS and SHBG by immunoassay in matched pairs of serum samples collected at baseline (D0) and at peak serum progestin levels at 25 weeks post initiation (25W) (n = 214–218 pairs). Both contraceptives substantially decreased, from D0 to 25W, the total testosterone [DMPA-IM D0 0.560, 25W 0.423 nmol/L, -24.3% (p < 0.0001); NET-EN D0 0.551, 25W 0.253 nmol/L, -54.1%, (p < 0.0001)], SHBG [DMPA-IM D0 45.0, 25W 32.7 nmol/L, -29.8% (p < 0.0001); NET-EN D0 50.2, 25W 17.6 nmol/L, -65.1% (p < 0.0001)], and calculated free testosterone levels [DMPA-IM D0 6.87, 25W 5.38 pmol/L, -17.2% (p = 0.0371);

**Data Availability Statement:** "The raw data is available without restriction and has been deposited in the BioStudies public database. The data is publicly available at https://www.ebi.ac.uk/biostudies/studies/S-BSST1403 or DOI 10.6019/S-BSST1403."

**Funding:** This work was supported by the U.S. National Institutes of Health and South African Medical Research Council through its U.S.-SA Program for Collaborative Biomedical Research (R01HD083026 [NICHD & NIAID] and R01AI152118 [NIAID]) to J.P.H. (https://www.nih.gov & https://www.samrc.ac.za), and a UCT Vice Chancellor's Advancing Womxn award to J.P.H (https://uct.ac.za). The clinical trial was funded by the South African Medical Research Council Grants, Innovation and Product Development (SAMRC N/A grant number) grant to M.S-M (https://www.samrc.ac.za/innovation/grants-innovation-and-product-development). The funders had no role in study design, data collection and analysis, decision to publish, or preparation of the manuscript.

**Competing interests:** The authors have declared that no competing interests exist.

NET-EN D0 6.00, 25W 3.70, -40.0% (p < 0.0001)]. After adjusting for change from D0, the total testosterone, SHBG and calculated free testosterone levels were significantly higher for DMPA-IM than NET-EN (64.9%, p < 0.0001; 101.2%, p < 0.0001; and 38.0%, p = 0.0120, respectively). The substantial and differential decrease in testosterone and SHBG levels does not explain our previous finding of no detected decrease in risky sexual behavior or sexual function for DMPA-IM or NET-EN users from D0 to 25W. Medroxyprogesterone (MPA) and NET are androgenic and are both present in molar excess over testosterone and SHBG concentrations at 25W. Any within or between contraceptive group androgenic effects on behavior in the brain are likely dominated by the androgenic activities of MPA and NET and not by the decreased endogenous testosterone levels. The clinical trial was registered with the Pan African Clinical Trials Registry (PACTR 202009758229976).

## Introduction

Progestin-only injectable contraceptives, mainly depo-medroxyprogesterone acetate intramuscular (DMPA-IM), are the most common contraceptive methods in sub-Saharan Africa [1–3], which has a high incidence and prevalence of HIV, particularly among young women and girls [4]. Meta-analyses of higher quality observational clinical data reported a significant 40–50% increased risk of HIV acquisition with DMPA-IM compared to no hormonal contraception, unlike for limited data on NET-EN [5, 6]. Head-to-head comparisons of HIV risk among women using DMPA-IM versus NET-EN indicated a potential 32–41% increase in HIV risk for DMPA-IM users versus NET-EN users [5, 7, 8]. The Evidence for Contraceptive Options and HIV Outcomes (ECHO) randomized trial comparing DMPA-IM, the copper intra uterine device (IUD) and Jadelle, a levonorgestrel (LNG)-containing implant, did not detect a significant difference in HIV acquisition risk of 50% or more between these methods [9]. However, the ECHO trial data do not inform on the risk for HIV infection for DMPA-IM compared to NET-EN, or for DMPA-IM compared to no contraception or irregular use of condoms [9]. Obtaining robust data on the effects of DMPA-IM and NET-EN on factors that potentially affect risk for HIV acquisition is important for understanding their risks and benefits and the biological mechanisms thereof. The Women's Health, Injectable Contraception and HIV (WHICH) trial, an open-label clinical trial that randomized women to DMPA-IM or NET-EN, investigated differences in hormonal, psychological, behavioral, menstrual and immune effects within and between the two contraceptives. While both contraceptives substantially and similarly reduced estradiol to postmenopausal levels one week after the 6-month injection, the data suggested more sexual exposure to HIV with DMPA-IM than NET-EN [10].

Levels of endogenous sex steroid hormones are likely to play a role in multiple physiological pathways and health outcomes, including susceptibility to sexually transmitted infections and HIV, immune function and sexual behavior [11]. While estradiol is known to be protective against HIV acquisition in the female genital tract [11], the role of androgens [12, 13] in HIV acquisition is unknown. Circulating levels of sex hormones reportedly modify cellular morphology in the brain [14] and influence higher brain functions such as cognition, memory and mood [15]. Indeed, decreased testosterone levels are associated with several undesired effects such as increased headaches, mood changes, and reduced sexual desire and libido [16]. Thus, decreased testosterone levels may decrease libido resulting in less exposure to HIV. Evidence on the effects of libido is, however, contentious as some women on combined oral contraception (COC) had decreased testosterone levels without a decrease in libido [17]. It is also

unclear whether sexual function in women is associated with endogenous androgen concentrations due to insufficient robust data and uncertainty relating to the sensitivity and specificity of androgen quantification assays in some studies [18]. In addition, hormonal contraception is also associated with several androgenic effects including acne, hirsutism, weight gain, androgenic alopecia, unfavourable lipid profiles, and diabetes [19–23]. DMPA-IM has been linked to increased incidence of type 2 diabetes, oily skin and acne in women [20, 22–24], while very little information is available on the androgenic effects of NET-EN in women. Furthermore, it is unknown to what extent any androgenic effects may be attributed to the known androgenic activity of MPA and NET via the androgen receptor (AR) *in vitro* [25].

Testosterone is one of the major androgens in the serum of premenopausal women [26, 27]. However, clinical studies investigating the effects of progestin-only contraceptives on testosterone levels are limited, and the available studies mainly assessed effects in women using COCs which result in reduced total serum testosterone concentrations (reviewed in [12]). Earlier studies showed decreased serum testosterone levels in postmenopausal breast cancer patients after administration of oral MPA [28]. Similarly, it has been shown that subcutaneous DMPA (DMPA-SC) [29] and the LNG implant [30] both decrease testosterone levels in premenopausal women. A decrease in testosterone was also detected in premenopausal women (3–4 women) and in a transgender population [31] administered DMPA-IM, while a significant decrease in testosterone was not detected in postpartum women administered NET-EN [32]. To understand the clinical significance of changes in testosterone levels in women, it is important to have accurate data on the effects of progestins not only on total testosterone levels, but also that of free testosterone and sex hormone binding globulin (SHBG) levels. Total testosterone includes both the biologically inactive, circulating, SHBG-bound testosterone, as well as the biologically active testosterone circulating either free (not bound to plasma protein) or that weakly bound to albumin [33]. Studies to date suggest that the effects of progestins on testosterone and SHBG are dependent on the type of contraceptive. For instance, ethinyl estradiol combined with either LNG or drospirenone (DRSP) reduced total and free testosterone levels, but increased SHBG levels [34]. This increase in SHBG levels is likely due to ethinyl estradiol as it is known to increase hepatic SHBG production (reviewed in [35]). In contrast, the progestin-only injectable DMPA-SC is associated with a significant decrease in total testosterone and SHBG, but not in free testosterone levels [29]. To our knowledge, only two other non-comparative studies have shown that the injectable contraceptives DMPA-IM [36] and NET-EN [37] decrease SHBG levels.

There is a lack of robust data on the effects of DMPA-IM and NET-EN on testosterone and SHBG levels, as well as their relative effects. Furthermore, no data are available for the effects of these contraceptives at peak serum progestin levels. In this study, we compared the serum levels of testosterone and SHBG, as well as changes in the levels of total and calculated free testosterone and SHBG, within and between two arms of the WHICH trial randomizing women to DMPA-IM or NET-EN, at peak progestin levels.

## Methods

### Primary study, ethics and biosafety

This study is a secondary study from the open-label randomized WHICH clinical trial. The primary aims of the trial were estradiol levels and menstrual, psychological and behavioral measures relevant to HIV risk. The WHICH study protocol and primary study have been reported elsewhere [10]. The study was registered retrospectively with the Pan African Clinical Trials Registry (PACTR 202009758229976 https://pactr.samrc.ac.za/Search.aspx). All women provided informed, written consent to authorize study participation and storage of samples.

The study adhered to the ethical principles outlined in the Declaration of Helsinki (World Medical Association, 2011) and the Constitution of the Republic of South Africa (Bill of Rights). Ethical approval for the secondary study conducted at the University of Cape Town (UCT) was obtained from the UCT Faculty of Health Sciences Human Research Ethics Committee (HREC REF no. 664/2018). The authors did not have access to information that could identify individual participants during or after data collection.

## Study design and sample collection

Briefly, HIV-negative young women (18–40 years) seeking contraception at the East London and Mdantsane public health clinics and hospitals (Frere and Cecilia Makiwane Hospitals), South Africa (331 participants), and the research site of the MatCH Research Unit (MRU), University of the Witwatersrand, based in Durban, KwaZulu-Natal, South Africa (189 participants) were randomized to 150 mg DMPA-IM 12-weekly or 200 mg NET-EN 8-weekly. Exclusion criteria were participants who received DMPA-IM in the previous 6 months or NET-EN in the previous 4 months, were living with HIV, or were using or intending to use medication which might have interfered with biological measurements such as steroids or drugs affecting renal function such as pre-exposure prophylaxis drugs (for HIV). Participants were recruited and followed from 5 November 2018 to 30 November 2019. We screened 546 and randomized 521 women to DMPA-IM (262) and NET-EN (259). A total of 86.9% (n = 453) completed a 25-week study visit with a similar number completing in both method groups.

Blood samples were collected at baseline (D0) and at 25 weeks (25W), i.e. about one week after the 6-month NET-EN (the 4th NET-EN injection) or DMPA-IM (the 3rd DMPA-IM) injection, and serum was separated and stored at -80°C.

## Total testosterone measurements

These measurements were performed between 2 January 2020 and 31 December 2022. Testosterone was quantified by ultra-high performance liquid tandem mass spectrometry (UHPLC-MS/MS) on stored serum samples from WHICH study participants at D0 and 25W, as described in S1 Appendix. Testosterone data were obtained for 214 and 215 participants, at D0 and 25W respectively, from the DMPA-IM arm, with D0 data for one participant being absent for technical reasons. Testosterone data were obtained for 219 and 218 participants at D0 and 25W, respectively from the NET-EN arm. Accuracy (% BIAS) and precision (% CV) were both less than 15% at all concentrations tested (0.05, 0.1, 0.5, 5 and 50 ng/mL) (S1 Table in S1 File). Recovery (% Extraction efficiency) and matrix effects were both within acceptable limits (S1 Table in S1 File). The limit of detection (LOD) and lower limit of quantification (LLOQ) for testosterone were 0.0250 ng/mL (0.0870 nmol/L) and 0.0500 ng/mL (0.173 nmol/L), respectively, while the upper limit of quantification (ULOQ) was 50.0 ng/mL (173.37 nmol/L) (S2 Table in S1 File). A linear calibration curve was obtained between the LLOQ and ULOQ ($R^2 > 0.996$). All researchers performing the assays were blinded.

## SHBG measurements and calculation of free testosterone

These measurements were performed between 2 January 2020 and 31 December 2020. SHBG was measured on stored serum samples from WHICH study participants at D0 and 25W by means of chemiluminescent microparticle immunoassay (CMIA) (Abbott Laboratories) (sensitivity level of quantification (LOQ) 0.02 nmol/L; no detectable cross-reactivity). All researchers performing the assays were blinded. Free testosterone was calculated according to the method of Vermeulen et al. [38]. SHBG data were obtained for 217 and 216 participants, at D0 and 25W respectively, from the DMPA-IM arm, with 25W data for one participant being

absent for technical reasons. SHBG data were obtained for 219 and 218 participants at D0 and 25W, respectively, from the NET-EN arm, with 25W data for one participant being absent for technical reasons. Free testosterone was calculated for all those participants for whom SHBG and testosterone data was available (214 and 215 for D0 and 25W, respectively in DMPA-IM arm; 219 and 217 for D0 and 25W, respectively, for NET-EN arm).

## Data analysis

UHPLC-MS/MS data collection and analysis were performed using MassLynx 4.2 (Waters Corporation). The ratio of the analyte peak area to internal standard peak area was determined for all the calibration curve samples, internal quality controls (IQCs) and serum samples. Testosterone values below the LLOQ, but above the LOD were assigned 0.5 x LLOQ (n = 16 D0; n = 30 25W), while values lower than LOD were assigned as 0.000 (n = 5 25W). For SHBG all the values were above the level of quantification (LOQ) (0.02 nmol/L).

We performed a modified intention-to-treat (mITT) analysis on the whole cohort and a per protocol (PP) analysis on a subgroup of the whole cohort. To obtain a subgroup for the PP analysis, we used UHPLC-MS/MS data obtained for study (MPA and NET) and non-study (LNG, nestorone, etonogestrel and gestodene) progestin levels at D0 and 25W in donor-matched serum samples [39]. For the testosterone and SHBG subgroup PP analysis, we excluded results from all women (104/436 participants, or 23.8%) that had any non-study serum progestin at concentrations ≥ 1.5 nM, at either D0 or 25W, from the mITT group. In addition to the above-mentioned non-study progestins, for the PP analysis we also excluded women in the DMPA-IM arm that had NET concentrations greater than 1.5 nM, at either D0 or 25W, as well as women in the NET-EN arm that had MPA concentrations higher than 1.5 nM, at either D0 or 25W. In total 23.8% (104/346 participants) of the women were excluded in the PP analysis.

Results in Tables 2–4 and S3 Table in S1 File were analysed using Stata version 16 (College Station, TX: StataCorp LLC). For total testosterone and SHBG, D0 and 25W values were nmol/L, while pmol/L values for free testosterone were used. Shapiro-Wilks normality tests indicated that all hormonal data were not normally distributed, and hence data are expressed as median with interquartile range (IQR) (Table 2 and S3 Table in S1 File). A mixed-effects linear regression model was fitted for each of the (natural) log-transformed outcomes (total testosterone, SHBG, free testosterone). Random effects to account for repeated measures within participants and to account for clustering by site were included in the model. A model coefficient, $\beta$, on the log-scale can be back-transformed using $e^{\beta}$. To facilitate the interpretation of the results, we report the percentage changes on the original scale, calculated as $(e^{\beta} - 1) \times 100\%$. Thus, the mean differences between D0 and 25W as well as between allocated arms are presented as percentages (back-transformed coefficients) with 95% confidence intervals (CIs) (Tables 3 and 4). Fig 2 and S1 Fig in S1 File were generated using GraphPad Prism 9.31 from GraphPad Software, Inc. (La Jolla California, USA), while statistical differences determined by mixed-effects linear regression model, as mentioned above, are shown. All results were considered significant for $p < 0.05$.

## Results

### Primary data and baseline characteristics

Of 521 participants enrolled, results are reported for all matched serum samples available from the whole cohort for 435 (83%) participants both at baseline and at peak (mITT analysis). The trial profile is shown in Fig 1.The excluded participants in the mITT analysis include 11 (2%) that became HIV positive, 6 (1%) that became pregnant, 65 (12%) that were lost to follow up

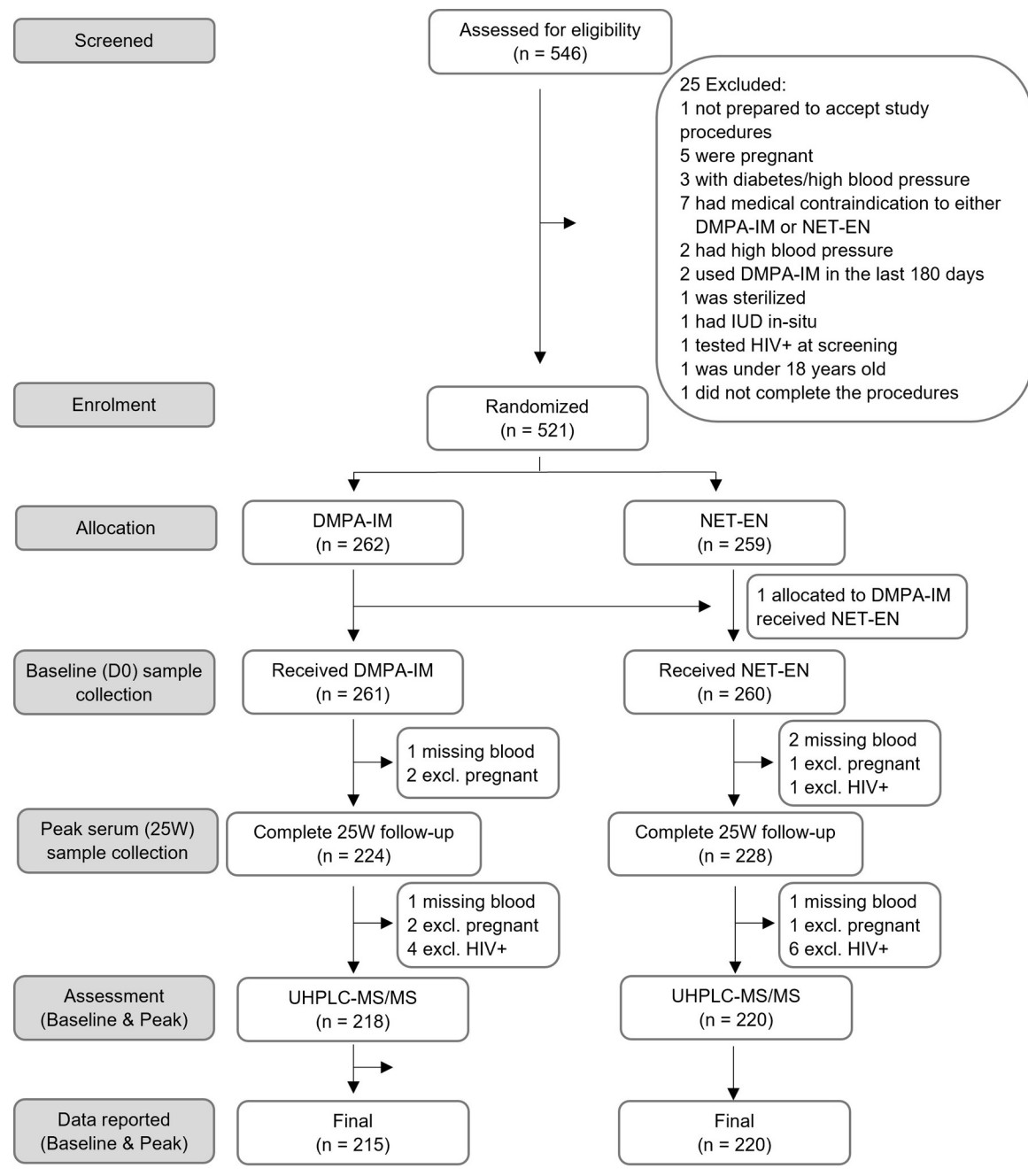

**Fig 1. Trial profile.**

(i.e., those that did not provide a 25W blood sample) and 5 participants that had a missing blood result for either SHBG or for testosterone, due to technical problems with the sample. Baseline data is shown in Table 1.

**Table 1. Baseline characteristics of women by randomization method$^\$$ (mITT analysis).**

| | Baseline (D0) | | | |
| | DMPA-IM | | NET-EN | |
| | | n | | n |
|---|---|---|---|---|
| **Age, years: Mean (SD)** | 25 (4.8) | 217 | 24.9 (4.7) | 219 |
| **Ethnicity** | | 217 | | 219 |
| Xhosa | 145 (66.8) | | 148 (67.6) | |
| Zulu | 67 (30.9) | | 71 (32.4) | |
| Mixed race | 1 (0.5) | | 0 (0.0) | |
| Other African ethnicity | 4 (1.8) | | 0 (0.0) | |
| **Previous use of method$^\#$** | | 217 | | 219 |
| DMPA-IM | 161 (74.2) | | 160 (73.1) | |
| NET-EN | 69 (31.8) | | 65 (29.7) | |
| **Marital status** | | 217 | | 219 |
| Single | 211 (97.2) | | 213 (97.3) | |
| Married | 6 (2.8) | | 6 (2.7) | |
| **Highest level of education** | | 217 | | 219 |
| Primary school, complete | 2 (0.9) | | 4 (1.8) | |
| High school, not complete | 88 (40.6) | | 76 (34.7) | |
| High school, complete | 82 (37.8) | | 95 (43.4) | |
| Post high school education | 45 (20.7) | | 44 (20.1) | |
| **Source of income** | | 217 | | 219 |
| Unemployed | 183 (84.3) | | 192 (87.7) | |
| Employed | 34 (15.7) | | 27 (12.3) | |

$^\$$ Unless indicated otherwise, values represent n-value (%)

$^\#$Prior to exclusion period, numbers given are for those that responded, and in brackets are % of those that responded. Note that some participants reported using both contraceptive methods prior to the exclusion period.

## DMPA-IM and NET-EN decrease total and free testosterone and SHBG concentrations

At baseline, the median total testosterone levels in the DMPA-IM and NET-EN arms were 0.560 nmol/L and 0.551 nmol/L, respectively (Table 2 and Fig 2). Both DMPA-IM and NET-EN significantly decreased total testosterone levels from D0 to 25W by 24.3% ($p < 0.0001$) and 54.1% ($p < 0.0001$), respectively (Table 3), with median concentrations at 25W in the DMPA-IM and NET-EN arms being 0.423 nmol/L and 0.253 nmol/L, respectively (Table 2). A significant difference in total testosterone levels between arms was detected at 25W with DMPA users having a 60.3% higher testosterone level than NET-EN users ($p < 0.0001$) (Table 4). No significant difference in total testosterone levels between arms was detected at D0 (Table 4). A significant difference was detected in total testosterone levels at 25W between arms, after adjusting for baseline, with DMPA-IM users showing a 64.9% ($p < 0.0001$) higher total testosterone level than NET-EN users (Table 4).

At baseline the median SHBG levels in the DMPA-IM and NET-EN arms were 45.0 nmol/L and 50.2 nmol/L, respectively (Table 2 and Fig 2). Both DMPA-IM and NET-EN significantly reduced SHBG levels from D0 to 25W by 29.8% ($p < 0.0001$) and 65.1% ($p < 0.0001$), respectively (Table 3), with median SHBG levels at 25W in the DMPA-IM and NET-EN arms being 32.7 nmol/L and 17.6 nmol/L, respectively (Table 2 and Fig 2). A significant difference in SHBG levels between arms was detected at 25W with DMPA users having an 89.9% higher

**Table 2. Total testosterone (nmol/L), SHBG (nmol/L) and free testosterone (pmol/L) outcomes at baseline and 25 weeks (mITT analysis).**

| | DMPA-IM | | NET-EN | |
|---|---|---|---|---|
| | Median (IQR) | n | Median (IQR) | n |
| **Total Testosterone (nmol/L)** | | | | |
| D0 | 0.560 (0.354; 0.815) | 214 | 0.551 (0.350; 0.853) | 219 |
| 25W | 0.423 (0.281; 0.610) | 215 | 0.253 (0.086; 0.385) | 218 |
| Change (25W - D0) | -0.119 (-0.288; 0.003) | | -0.291 (-0.513; -0.090) | |
| **SHBG (nmol/L)** | | | | |
| D0 | 45.0 (33.6; 68.8) | 217 | 50.2 (35.1; 72.4) | 219 |
| 25W | 32.7 (24.8; 44.4) | 216 | 17.6 (12.2; 22.9) | 218 |
| Change (25W - D0) | -12.2 (-25.1; -2.3) | | -32.3 (-51.5; -17.7) | |
| **Free Testosterone (pmol/L)** | | | | |
| D0 | 6.87 (2.81; 13.82) | 214 | 6.00 (2.00; 14.0) | 219 |
| 25W | 5.38 (2.28; 11.73) | 215 | 3.70 (0.87; 9.33) | 217 |
| Change (25W - D0) | -0.71 (-5.48; 1.33) | | -1.50 (-7.02; 1.16) | |

IQR (25th and 75th Percentile)

SHBG level than NET-EN users ($p < 0.0001$) (Table 4). At D0 no significant difference in SHBG levels was detected between the two arms (Table 4). After adjusting for the change from D0, DMPA-IM users had 101.2% ($p < 0.0001$) higher SHBG levels than NET-EN users at 25W (Table 4).

Median calculated free testosterone concentrations at baseline in the DMPA-IM and NET-EN arm were 6.87 pmol/L and 6.00 pmol/L, respectively (Fig 2 and Table 2). At 25W the median free testosterone levels in the DMPA-IM and NET-EN arms were 5.38 pmol/L and 3.70 pmol/L, respectively (Table 2). Both DMPA-IM and NET-EN significantly reduced free testosterone levels from D0 to 25W by 17.2% ($p = 0.0371$) and 40.0% ($p < 0.0001$), respectively (Table 3). Additionally, a significant difference in free testosterone levels between arms was detected at 25W with DMPA users having a 37.7% higher free testosterone level than NET-EN users ($p = 0.0110$) (Table 4). At D0 no significant difference in free testosterone levels was detected between the two arms (Table 4). After adjusting for the change from D0, DMPA-IM users had 38.0% ($p = 0.0120$) higher free testosterone levels at 25W than NET-EN users (Table 4).

In a subgroup PP analysis, after excluding for non-study progestins, significant differences were detected for the same comparisons for the whole cohort (mITT analysis) compared to the subgroup (PP analysis), for total testosterone, SHBG and free testosterone concentrations (S3 Table and S1 Fig in S1 File). The baseline characteristics for the subgroup of women randomized to DMPA-IM or NET-EN used in the PP analysis are shown in S4 Table in S1 File.

## Discussion

We report for the first time on the effects of DMPA-IM and NET-EN on total and free testosterone and SHBG levels at peak progestin levels from a randomized trial. We detected substantial decreases from D0 to 25W in measured total testosterone (-24.3% and -54.1%) and SHBG levels (-29.8% and -65.1%) and calculated free testosterone levels (-17.2% and -40.0%), for DMPA-IM and NET-EN users, respectively. Whether the lower median concentrations of total testosterone at 25W could be classified as hypoandrogenic or post-menopausal, is unclear from the literature [27, 40, 41]. When comparing the mean percentage difference in change

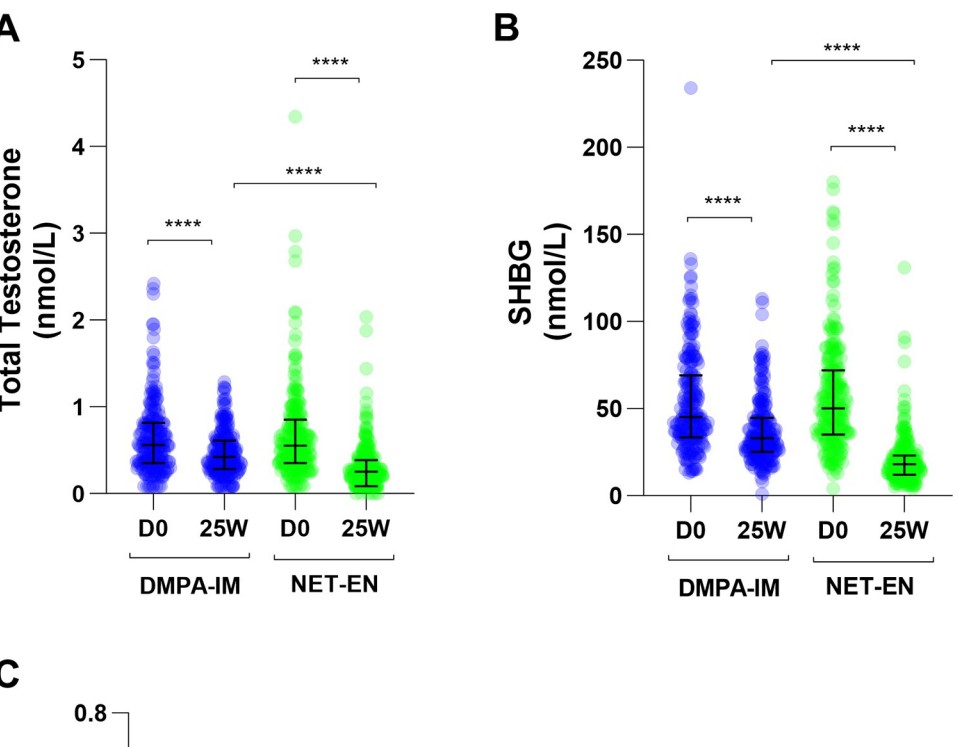

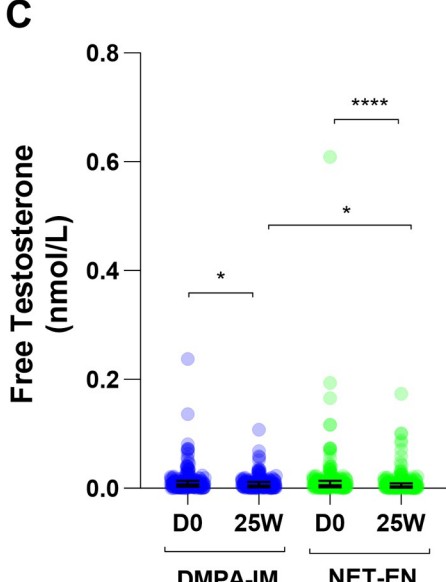

**Fig 2. Total testosterone (nmol/L), SHBG (nmol/L) and free testosterone (nmol/L), outcomes at baseline and 25 weeks (mITT analysis).** Graphs indicate median with interquartile range (IQR). Significant differences were calculated by mixed effects linear regression, accounting for repeated measurements per participant and clustering by site and are indicated by asterisks where * and **** represent p<0.05 and p<0.0001, respectively.

between these contraceptives from D0 to 25W, we report that DMPA-IM use results in a significantly smaller change in total testosterone, SHBG levels and free testosterone than NET-EN use.

To our knowledge there is no published literature to allow direct comparison between our results for DMPA-IM and NET-EN at peak progestin concentrations. One observational study reported values for DMPA-IM that are 27-fold greater and 2.5-fold lower [42] than the values we determined for total and free testosterone, respectively. A study among 15 women using

**Table 3. Mean percentage changes (95% CI) in total testosterone, SHBG and free testosterone (mITT analysis).**

| | DMPA-IM | NET-EN | DMPA-IM vs NET-EN |
|---|---|---|---|
| | Mean (95% CI)* | Mean (95% CI)* | p-value* |
| **Total Testosterone** | | | |
| % Change from D0 | -24.3 (-30.5; -17.5) | -54.1 (-57.9; -50.0) | **<0.0001** |
| **SHBG** | | | |
| % Change from D0 | -29.8 (-34.6; -24.7) | -65.1 (-67.5; -62.6) | **<0.0001** |
| **Free Testosterone** | | | |
| % Change from D0 | -17.2 (-30.7; -1.1) | -40.0 (-49.8; -28.4) | **0.0120** |

* estimates and p-values from mixed effects linear regression, accounting for repeated measurements per participant and clustering by site. Note, mixed effects linear regression indicated that both DMPA-IM and NET-EN significantly decreased total testosterone (p < 0.0001 for both), SHBG (p < 0.0001 for both) and free testosterone (DMPA-IM p = 0.0371; NET-EN p < 0.0001) levels from D0 to 25W.

the 104 mg DMPA-SC injection reported a significant decrease in total testosterone and SHBG, measured by immunoassay, but not for calculated free testosterone, one week after the 3-month DMPA-SC injection [29]. Their values are about 2–4-fold higher than our median total testosterone values, but comparable to our median SHBG values and about 5–6-fold lower than our values calculated for free testosterone. Very little information is available for NET-EN, but one study reported a reduction in SHBG levels five days after NET-EN injection, with levels of SHBG 2.3-fold higher than our value at 25W [37]. Possible reasons for differences between our results and other results for DMPA-IM or NET-EN include lower power due to small sample sizes, confounding factors due to non-randomization, and differences in sampling times and methods of testosterone quantification. Our use of UHPLC-MS/MS is likely to have generated more accurate and lower values for testosterone than immunoassay methods [43, 44]. Nevertheless, our results, reviewed together with limited published data [45], suggest that most progestin-only contraceptives significantly and substantially decrease SHBG levels, and that DMPA-IM decreases total and calculated free testosterone levels to a greater extent than DMPA-SC at peak serum MPA levels, while both have and similar effects on SHBG levels.

The biological, behavioral and clinical consequences of substantially decreased levels of total and free serum testosterone and SHBG for both DMPA-IM and NET-EN are difficult to interpret. When comparing effects on sexual behavior between DMPA-IM and NET-EN, we have reported more sexual activity and more risky sexual behavior and possibly more exposure to HIV with DMPA-IM than NET-EN [10]. This is consistent with a greater decrease in total testosterone levels for NET-EN relative to DMPA-IM, as testosterone may be associated with increased libido. However, while the relative effects of these contraceptives are important, the

**Table 4. Pairwise comparisons of hormonal results for DMPA-IM vs NET-EN expressed as mean percentage differences (95% CI) (mITT analysis).**

| | Mean percentage difference between DMPA-IM and NET-EN at D0 | | Mean percentage difference between DMPA-IM and NET-EN at 25W | | Mean percentage difference at 25W between DMPA-IM and NET-EN, after adjusting for baseline | |
|---|---|---|---|---|---|---|
| | DMPA$_{D0}$ –NET-EN$_{D0}$ | | DMPA$_{25W}$ –NET-EN$_{25W}$ | | | |
| | Mean % (95% CI)* | p-value* | Mean % (95% CI)* | p-value* | Mean % (95% CI)* | p-value* |
| **Total Testosterone** | -2.9 (-13.6; 9.3) | 0.6347 | 60.3 (42.4; 80.4) | <0.0001 | 64.9 (46.1; 86.1) | <0.0001 |
| **SHBG** | -5.6 (-14.6; 4.3) | 0.2556 | 89.9 (71.8; 109.9) | <0.0001 | 101.2 (82.2; 122.3) | <0.0001 |
| **Free testosterone** | -0.2 (-21.9; 27.5) | 0.9867 | 37.7 (7.6; 76.2) | 0.0110 | 38.0 (7.3; 77.4) | 0.0120 |

*estimates and p-values from mixed effects linear regression, accounting for repeated measurements per participant and clustering by site.

individual effects from baseline to 25W are also biologically and clinically relevant. If increased testosterone results in increased sexual activity in women, one would expect a decrease in sexual behavior within both arms. However, a substantial decrease in sexual behavior from baseline to 25W was not detected in the WHICH cohort for either contraceptive [10].

An important potential confounding factor in understanding the effects of DMPA-IM and NET-EN on androgenic effects is that both these progestins are themselves androgenic. They have similar binding affinities for and potencies via the AR *in vitro* when compared head-to-head to each other and dihydrotestosterone [25, 46], although one study reported that MPA is less potent than testosterone and dihydrotestosterone [47]. While some studies have reported that NET has more androgenic activity than MPA in animal pre-clinical models [48], there is no robust data on androgenic activity of DMPA-IM and NET-EN in women. It is not possible to extrapolate these *in vitro* and animal data directly to relative androgenic activities in women due to multiple confounding factors, including metabolism, cross-talk with other pathways and species-, gene- and cell-specific effects. We have reported that the medium peak serum concentrations for MPA and NET in DMPA-IM and NET-EN users are 6.6 and 14 nmol/L, respectively, in these WHICH trial samples [39]. Notably, these concentrations are much higher than those for endogenous total or calculated free testosterone [DMPA-IM 0.423 nmol/L and 5.38 pmol/L; NET-EN 0.253 nmol/L and 3.70 pmol/L, respectively] as well as those of estradiol (25W DMPA-IM 77 pmol/L; 25W NET-EN 70 pmol/L) [10]. Thus, it is possible that any androgenic effects of DMPA-IM and NET-EN are dominated by the androgenic properties of MPA and NET themselves, rather than the relatively very low but differential levels of testosterone reported here, and postmenopausal but similar levels of estradiol for both contraceptives [10].

One potential confounding factor in our randomized study is misreporting of DMPA-IM or NET-EN use, or use of contraceptives containing estrogens such as COCs, before and/or during the trial. We and others have reported that it is common for women not to self-report non-study progestin use before initiation and during clinical trials on contraception [39, 49]. We performed a PP analysis of the testosterone and SHBG data on a subgroup of participants after excluding those with concentrations of non-study progestins above 1.5 nM at either D0 or 25W. These PP analyses resulted in the same significant differences compared to the mITT results, and comparable median values for total and free testosterone and SHBG, suggesting that non-study progestin use did not introduce significant bias to these values. Our testosterone and SHBG findings are unlikely to be confounded by differences in baseline characteristics between contraceptive arms, given the stringent randomization process.

## Conclusions

Total and free testosterone and SHBG potentially affect multiple physiological pathways and clinical outcomes. Our published findings of increased risky sexual behavior for DMPA-IM relative to NET-EN users are consistent with the differential decrease in total endogenous testosterone levels. However, our lack of detection of a general decrease in risky sexual behavior from D0 to 25W for either contraceptive is not consistent with a substantial decrease in endogenous testosterone levels being the major determinant of changes in sexual behavior. Taken together, it is likely that progestins themselves are the major determinants of androgenic effects, including on sexual behavior in the brain, that potentially affect HIV acquisition risk for DMPA-IM and NET-EN, as well as their relative effects. Understanding these complex mechanisms requires more research.

## Supporting information

**S1 File. S1 Fig and S1 to S4 Tables.**
(DOCX)

**S1 Appendix. Appendix 1 testosterone quantification.**
(DOCX)

## Acknowledgments

We thank all the women who participated in the WHICH clinical trial. We thank Karen van der Merwe for project administration and assistance in preparation and submission of the manuscript. We thank Marietjie Stander and Erick van Schalkwyk for assistance with the UHPLC-MS/MS assays.

## Author Contributions

**Conceptualization:** Chanel Avenant, Mandisa Singata-Madliki, George J. Hofmeyr, Janet P. Hapgood.

**Data curation:** Chanel Avenant, Yusentha Balakrishna, Ishen Seocharan.

**Formal analysis:** Chanel Avenant, Yusentha Balakrishna.

**Funding acquisition:** Mandisa Singata-Madliki, Janet P. Hapgood.

**Investigation:** Chanel Avenant, Alexis J. Bick, Johnson M. Moliki, Sigcinile Dlamini.

**Methodology:** Chanel Avenant, Karl-Heinz Storbeck, Salndave Skosana, Janet P. Hapgood.

**Project administration:** Chanel Avenant, Janet P. Hapgood.

**Resources:** Mandisa Singata-Madliki, Jenni Smit, Mags Beksinska, Ivana Beesham, Joanne Batting.

**Supervision:** Chanel Avenant, Janet P. Hapgood.

**Visualization:** Chanel Avenant.

**Writing – original draft:** Chanel Avenant, Donita Africander, George J. Hofmeyr, Janet P. Hapgood.

**Writing – review & editing:** Chanel Avenant, Mandisa Singata-Madliki, Alexis J. Bick, Donita Africander, Yusentha Balakrishna, Karl-Heinz Storbeck, Johnson M. Moliki, Sigcinile Dlamini, Jenni Smit, Mags Beksinska, Ivana Beesham, Joanne Batting, George J. Hofmeyr, Janet P. Hapgood.

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
