## [Decision Letter · Decision Letter 0]

9 Feb 2024

PONE-D-23-20153The injectable contraceptives depot medroxyprogesterone acetate and norethisterone enanthate substantially and differentially decrease testosterone and sex hormone binding globulin levels: a secondary study from the WHICH randomized clinical trial.PLOS ONE

Dear Dr. Hapgood,

Thank you for submitting your manuscript to PLOS ONE. After careful consideration, we feel that it has merit but does not fully meet PLOS ONE’s publication criteria as it currently stands. Therefore, we invite you to submit a revised version of the manuscript that addresses the points raised during the review process.

We look forward to receiving your revised manuscript.

Kind regards,

Renee Ridzon

Academic Editor

PLOS ONE

Journal Requirements:

“This work was supported by the U.S. National Institutes of Health and South African Medical Research Council through its U.S.-SA Program for Collaborative Biomedical Research (R01HD083026 [NICHD & NIAID] and R01AI152118 [NIAID]) to J.P.H. (https://www.nih.gov & https://www.samrc.ac.za), and a UCT Vice Chancellor’s Advancing Womxn award to J.P.H (https://uct.ac.za). The clinical trial was funded by the South African Medical Research Council Grants, Innovation and Product Development (SAMRC N/A grant number) grant to M.S-M (https://www.samrc.ac.za/innovation/grants-innovation-and-product-development). The funders had no role in study design, data collection and analysis, decision to publish, or preparation of the manuscript.”

Additional Editor Comments (if provided):

Thank you for your submission and please see reviews below. It has been exceedingly difficult to find reviewers so apologize for delay in return of this and in fact there were invitations to over 25 people to act as a reviewer for this manuscript. This is a substudy of measurements of hormones from the larger WHICH study. There is reference to findings of the larger study multiple times and the results of this study are helpful to put context around the data presented here. Unfortunately, the references are designated as "submitted" so the reader is not able to see that study. This led me to wonder if it would be better to publish this one after the publication of the main study. Given the time that has passed, this main study may be slated for publication which would be ideal.

Lines 54-55 Will the reader understand what the abbreviations D0 and 25W mean? Would it be clearer to say in the beginning of the sentence that there was a substantial decrease from baseline to week 25 (if I am understanding what these abbreviations mean) and then give the % decline in the values. It appears that the decrease is calculated as 25W-D0 and the decrease is expressed as a negative %. A negative decrease (seems like a double negative) could be interpreted as an increase. If not, would be clearer to say that the decrease is a % (without the negative sign) or that the value at 25W is x% lower than D0? This comment pertains to the results section as well. If this is the standard way this information is expressed, please disregard the comment.

Line 59-The % difference (decrease) for testosterone and SHBG is expressed without a leading negative sign. Why is the difference expressed as a positive number here and negative above? Is there a reason why both a decrease in median level and a mean percentage difference is given?

Line 64-The time period of measurement expressed here appears to be D0 to 24W whereas above it is D0 and 25W. Is this correct? Why are these times different and why the is the formatting different for the presentation of the number of weeks (24W vs W25)?

Line 119- Is the implication here that decreased levels of testosterone impact susceptibility to HIV though a decreased libido that leads to less exposure or is it another mechanism? Should this be explained further here?

Line 152-It might be interesting to tell the reader whether it is the bound or the free testosterone that is active (or the different activities of these 2 moieties), if there is room, especially as it is later stated how the different hormones impact, free, bound and total testosterone.

Line 188-Presumably this is an open label study due the difference in the dosing interval of the agents. It would be helpful to state this.

Line 196-Are the PrEP drugs referred to here for HIV? That should be stated.

Line 258-It is stated that a subgroup was used for the PP analysis and that this group excluded of persons who used “non-study progestin >1.5”. It is very unclear what this is and is it possible that examples be given?

Line 298-Why are the numbers 219 and 217 listed in the white lines of this table under “previous use of method” whereas in the other parts of this table these totals are in the gray lines? “Previous use of method” should be reworded to “Previous method used”. Since the numbers for this category add to greater than the total for each arm, need to tell the reader that some participants used both of the methods listed. Why are Single and Married listed under this heading?

Line 306-Usually, the p value is listed after the 2 values being compared.

Line 342-Perhaps state for clarity “DMPA-IM and NET-EN significantly reduced median SHBG levels from baseline to 25W by -27% and -66%, respectively (Table 3)". Is there a reason that the levels are presented in an inconsistent manner here? For example, the SHBG levels are presented as values in nmol/L with a difference expressed in concentration in nmol/L but the SHBG is presented as a negative % with the difference presented as % change.

Line 347-“Median free testosterone concentrations at baseline and 25W were not significantly different between the two arms at baseline (p = 0.842)” Does this mean that all 4 of the measurements were not different from each other? Or does that mean there was not a difference between baseline and 25W in each of the 2 arms? Please clarify the statement. The statement seems to contradict the next sentence of “At 25W the 349 median free testosterone level in the DMPA-IM arm of 5.00 pmol/L was significantly 350 higher than 4.00 pmol/L in the NET-EN arm (p = 0.0124)”. This entire paragraph is hard to followed and needs to be rewritten for clarity.

Line-358-Does data need to be presented for both medians and means? If so why? Since it is stated that results were similar for medians and means, it is not clear if this information is needed.

Line 374-The discussion seems a bit long and could be shortened with the discussion on other studies made more terse.

Line 376-It is stated “significant decreases from D0 to 25W in measured total testosterone (-24% and -54%)”. The decrease is a percent, not a negative percent. By saying that there is a decrease from baseline to 25W, it is understood that the value is lower, the % expresses the magnitude of the decrease. Adding the negative sign seems to make this confusing.

Line 431-Does 4/5 and 3/5 mentioned here refer to markers of sexual behavior? If so, perhaps reword to “no significant differences were detected for 4/5 markers in 432 the DMPA-IM arm and for 3/5 markers in the NET-EN arm”.

Line 437-Sometimes HIV is used and sometimes HIV-1 is used, as here. Is there a reason for this difference?

Line 438-Data on sexual desire is not included in this paper and the discussion should be focused on the data from this study. That there was a difference in sexual desire is noted in the WHICH cohort is mentioned on Line 429 and that seems sufficient and the discussion here should focus on this paper’s data, especially since other paper is not yet published.

Line 444-The statistical power of the WHICH study is not the subject of this discussion and no information has been given to the reader to support claims about the power of the study on sexual behavior results.

Line 460-Can it be stated that data have been reported when the paper has not yet been published?

Line 481-What is the basis for saying that the study is high-powered? No information on the power of the study is given in the methods. How could there be misreporting of DMPA-IM or NET-EN use when these are the interventions that are being examined in this study and presumably being administered in a controlled manner?

Line 490-stated “we would have had to exclude a large proportion of the women (for use of non-study progestins) and substantially reduced the power of this study” What are these and how many of the participants used them? Was it possible to exclude such women from the study? If not, why not?

Line 506-“Our findings of increased risky sexual behavior for DMPA-IM relative to NET-EN users” no data on this has been presented and this should not be part of the conclusions of this study. Further is stated that it is likely that progestins are determinants of androgenic effects. Given that it is also stated that a large number of the women in this study may have used non-study progestins, does this introduce significant bias?

Reviewers' comments:

Reviewer's Responses to Questions

**Comments to the Author**

1. Is the manuscript technically sound, and do the data support the conclusions?

Reviewer #1: No

Reviewer #2: Yes

2. Has the statistical analysis been performed appropriately and rigorously? 

Reviewer #1: No

Reviewer #2: Yes

3. Have the authors made all data underlying the findings in their manuscript fully available?

Reviewer #1: Yes

Reviewer #2: Yes

4. Is the manuscript presented in an intelligible fashion and written in standard English?

Reviewer #1: No

Reviewer #2: Yes

5. Review Comments to the Author

Reviewer #1: General comments:

This manuscript reports a secondary analysis from a clinical trial that explored the differences in serum testosterone and sex hormone binding globulin levels between two injectable contraceptives.

My main concern regards the statistical methods. The methods seem to lack coherence. For instance, I found it confusing why the authors claim non-parametric tests are required for all data, but then use methods that assume normality. There may also be methods used that are not reported in the methods section.

Also, I will point out that the choice of running a statistical model is more about whether the residuals are normally distributed and not the outcome itself. Linear models are somewhat robust against deviations from normality. Looking at figure 2, I think analyzing a log-transformed outcomes is feasible. I would much prefer if you were able to fit a single statistical model for each of outcome and then evaluate differences between groups and time points. That would reduce the risk of type I errors and allow for much more coherent methods and results sections.

Although I am here to evaluate design and methods, I strongly suggest reducing the introduction and probably the discussion. In the vast majority of papers, the introduction can be 3-4 paragraphs, i.e., (1) broad background on problem, (2) specifics on problem to be addressed, and (3) objectives and brief synopsis of design and methods.

Thus, I don't feel the paper presently meets criterion 3 (Experiments, statistics, and other analyses are performed to a high technical standard and are described in sufficient detail) and criterion 5 (The article is presented in an intelligible fashion and is written in standard English) since it's wordy. Though, I think you could fix these.

Specific comments:

1. (lines 51-52) If you are using paired tests, this should be the sample size of the pairs.

2. (lines 54-60) Confidence intervals or p-values are needed for all of these comparisons.

3. (lines 225-226) This must be described somewhere in the manuscript especially since the information has not been published. There is no word count for PLOS ONE, so you should feel free to include this information here.

4. In lines 267-269, you note that all data are not normally distributed but then in lines 274-277 you use a model that assumes normality. Granted, you log-transformed your outcomes for the mixed models. Why can't that be done for the simple tests as well?

5. (Table 1, line 295, line 306, …) Claiming "similarity" is incorrect when using superiority tests. These statements suggest that the absence of evidence indicates evidence of absence. You can claim no difference, but not similarity. Statements like this need to be revised since a p>0.05 doesn't necessarily mean there were no differences. Significance testing for baseline imbalance in randomized trials been has regarded as unnecessary (see Altman DG. Comparability of randomised groups. The Statistician 1985; 125-136; Senn, S. Testing for baseline balance in clinical trials. Statistics in Medicine 1994; 1715-1726). I recommend removing the significance testing entirely from table 1.

6. (Table 3, line 315) I don't recall something in the methods describing analyses of percentage change.

7. (Table 4) I encourage you to report your effects on the scale in which they are analyzed.

8. In addition, you use the word "similar" in other situations, e.g., lines 363 and 366. Similar is vague and subjective and I suggest being specific about the comparisons made in these statements.

Reviewer #2: thanks for the opportunity to review this well written paper

A few minor comments only

Line 103 – please clarify what you mean by the “6 month injection” as neither of these are 6 month injections.

Please explain line 105 further

Line 196 – assume you mean PrEP for HIV - clarify

6. PLOS authors have the option to publish the peer review history of their article (what does this mean?). If published, this will include your full peer review and any attached files.

Reviewer #1: No

Reviewer #2: **Yes: **Katherine Gill

---

## [Author Response · Author response to Decision Letter 0]

23 Apr 2024

Thanks to the editor and the reviewers for all the very constructive suggestions and queries. We have attended to all the comments. 

A detailed point-by-point response can be found in the uploaded file "response to reviewers".

---

## [Decision Letter · Decision Letter 1]

16 May 2024

PONE-D-23-20153R1The injectable contraceptives depot medroxyprogesterone acetate and norethisterone enanthate substantially and differentially decrease testosterone and sex hormone binding globulin levels: a secondary study from the WHICH randomized clinical trial.PLOS ONE

Dear Dr. Hapgood,

Thank you for submitting your manuscript to PLOS ONE. Upon re-review, one referee has some additional comments that need to be addressed. Therefore, we invite you to submit a revised version of the manuscript that addresses the points raised during the review process.thank you for your attention to this Please ensure that your decision is justified on PLOS ONE’s publication criteria and not, for example, on novelty or perceived impact.

We look forward to receiving your revised manuscript.

Kind regards,

Renee Ridzon

Academic Editor

PLOS ONE

Additional Editor Comments:

Reviewers' comments:

Reviewer's Responses to Questions

**Comments to the Author**

1. If the authors have adequately addressed your comments raised in a previous round of review and you feel that this manuscript is now acceptable for publication, you may indicate that here to bypass the “Comments to the Author” section, enter your conflict of interest statement in the “Confidential to Editor” section, and submit your "Accept" recommendation.

Reviewer #1: (No Response)

2. Is the manuscript technically sound, and do the data support the conclusions?

Reviewer #1: Partly

3. Has the statistical analysis been performed appropriately and rigorously? 

Reviewer #1: No

4. Have the authors made all data underlying the findings in their manuscript fully available?

Reviewer #1: Yes

5. Is the manuscript presented in an intelligible fashion and written in standard English?

Reviewer #1: Yes

6. Review Comments to the Author

Reviewer #1: Thank you for considering my comments to the initial draft. To follow up on a couple of these comments:

Significance testing in table 1:

The main point of my comment is to provide a more accurate assessment of baseline imbalance. It is not about transparency. For example, two groups with small sample sizes will likely have p>0.05 for all characteristics. So, then having a smaller sample size is better…? No, it's because p-values are a poor assessment of baseline comparability since the p-value is driven by the sample size. Other methods, such as standardized mean differences, provide a better assessment of differences. I still believe this should be changed.

Normality assumption:

I'm fine with reporting medians and IQRs for summary statistics, but in Table 2 you are reporting medians and IQRs right next to p-values from regression models that assume normality. This remains confusing for me and I think has the potential for misinterpretation. If someone takes a quick glance at this, I fear s/he will either (1) think the summary statistics are means and 95% CIs or, more likely, think the p-value are for a difference in medians. I encourage you to report what is actually being tested next to the p-value to avoid this potential for confusion.

Additional comments:

- Please provide details on the back-transformation mentioned in line 258 since this is reported in the text and a table.

- (Table 3) For free testosterone, why is the DMPA-IM vs. NET-EN p-value < 0.05 when the two 95% CIs overlap?

(lines 299-301 and Table 4) It's not clear to me what the values in this table mean. For total testosterone, is the -2.9 a difference at baseline? Or is it a mean percentage change? It's not clear from the title or the notes what these values are.

7. PLOS authors have the option to publish the peer review history of their article (what does this mean?). If published, this will include your full peer review and any attached files.

Reviewer #1: No

---

## [Author Response · Author response to Decision Letter 1]

29 Jun 2024

Thank you for the comments.

The response is in the file "response to reviewers"

---

## [Editor Report · Decision Letter 2]

11 Jul 2024

The injectable contraceptives depot medroxyprogesterone acetate and norethisterone enanthate substantially and differentially decrease testosterone and sex hormone binding globulin levels: a secondary study from the WHICH randomized clinical trial.

PONE-D-23-20153R2

Dear Dr. Hapgood,

We’re pleased to inform you that your manuscript has been judged scientifically suitable for publication and will be formally accepted for publication once it meets all outstanding technical requirements.

Kind regards,

Renee Ridzon

Academic Editor

PLOS ONE
---

## [Editor Report · Acceptance letter]

15 Jul 2024

PONE-D-23-20153R2 

PLOS ONE

Dear Dr. Hapgood, 

I'm pleased to inform you that your manuscript has been deemed suitable for publication in PLOS ONE. Congratulations! Your manuscript is now being handed over to our production team.

Kind regards, 

on behalf of

Dr. Renee Ridzon 

Academic Editor

PLOS ONE